# Learning to Cycle: Body Composition and Balance Challenges in Balance Bikes Versus Training Wheels

**DOI:** 10.3390/jfmk10010053

**Published:** 2025-01-31

**Authors:** Cristiana Mercê, David Catela, Rita Cordovil, Mafalda Bernardino, Marco Branco

**Affiliations:** 1Sport Sciences School of Rio Maior, Santarém Polytechnic University, Avenue Dr. Mário Soares No. 110, 2040-413 Rio Maior, Portugal; catela@esdrm.ipsantarem.pt (D.C.); mafalda.bernardino@esdrm.ipsantarem.pt (M.B.); marcobranco@esdrm.ipsantarem.pt (M.B.); 2Sport Physical Activity and Health Research & Innovation Center (SPRINT), Santarém Polytechnic University, Complex Andaluz, Apart 279, 2001-904 Santarém, Portugal; 3Physical Activity and Health—Life Quality Research Centre (CIEQV), Polytechnique University of Santarém, Complex Andaluz, Apart 279, 2001-904 Santarém, Portugal; 4Interdisciplinary Center for the Study of Human Performance (CIPER), Faculty of Human Kinetics, University of Lisbon, Cruz Quebrada-Dafundo, 1499-002 Lisboa, Portugal; cordovil.rita@gmail.com; 5Quality Education—Life Quality Research Centre (CIEQV), Santarém Polytechnique University, Complex Andaluz, Apart 279, 2001-904 Santarém, Portugal

**Keywords:** learning, early childhood, motor skills, cycling milestone, independent cycling, health, exercise, fitness, body composition, balance

## Abstract

**Background/Objectives:** Empowering our children and youth to cycle empowers them to pursue a healthier, fuller, and more responsible life. The present study implemented the Learning to Cycle program with the following aims: (i) to promote learning to cycle; (ii) to investigate and compare the use of different learning bicycles, i.e., balance bicycle (BB) and bicycle with training wheels (BTW); (iii) to investigate the influence of body composition during this learning process. **Methods:** The program was implemented through a quasi-experimental study involving two intervention groups, with pre- and post-test evaluations. The program was applied to 50 children (M = 5.82 ± 0.94 years, 23 girls) who did not know how to cycle previously. One group explored the BB and the other the BTW for six sessions, followed by four more sessions with the conventional bicycle (CB) for both groups. The assessment of independent cycling was considered as the ability to perform, sequentially and unaided, and the various cycling milestones: self-launch, ride, and brake. The children’s body composition was accessed by the BMI’s percentile and classification according to their age and sex. **Results:** The program had a success rate of 88.24% for acquiring independent cycling, with 100% success in the BB group and 76.92% in the BTW group. The BB children learned significantly faster to self-launch, ride, brake, and cycle independently. Children with higher BMI percentiles faced greater challenges in achieving balance milestones. **Conclusions:** BB are recommended, especially for overweight and obese children, as they help develop balance from the onset, and showed to be more efficient in learning to cycle than the BTW.

## 1. Introduction

The importance of the bicycle transcends the issue of mobility, for which it was originally created. Riding a bicycle has recently been recognized as a foundational motor skill, which means that cycling promotes physical activity engagement and practice, as well as positive health trajectories throughout life [1]. Cycling is no longer just a matter of mobility, but also a matter of health. This active mode of transportation provides numerous health benefits across age groups, such as improved cardiorespiratory fitness [2,3], body composition [4,5], decreased risk of metabolic syndrome [6], and of overall mortality [7]. Nevertheless, cycling is not only about promoting physical health, but also about mental and social well-being. Riding a bicycle promotes moments of socialization and fun, develops relational and emotional skills [8,9], and also facilitates the environment exploration [10]. Moreover, cycling is not only about health, but also about culture. This means of transportation enables the (re)discovery of places and people [11], allowing children and young people to become more independent [10]. The scope of cycling does not end here, as cycling is also about protecting the environment and protecting our future. As an environmentally friendly means of transportation, choosing the bicycle over a polluting vehicle contributes to a better environment [12]. The sooner our children and youth start cycling, the sooner they will become familiar with this active mode of transportation and be able to enjoy all its benefits, as well as engaging in healthy life trajectories [1]. To this end, it is necessary to promote learning to cycle as early as possible, either in school context or outside of it.

Promoting early independent cycling involves understanding the best methods for children to learn how to ride a bicycle. Several studies have investigated the constraints involved in learning to cycle (e.g., [13,14,15,16,17]). All motor learning is influenced by several types of constraints [18], namely individual, i.e., a person’s characteristics such as body composition or motor experience; task, i.e., manipulable factors, such as the conditions for motor practice, such as equipment, organization, opportunities; and environment, i.e., factors not directly controllable, like the laws in nature, the rules of a culture, or the historical period in which we grow up. In this sense, a systematic review was recently conducted to identify the best strategies, i.e., constraint manipulations, to promote a more efficient learning to cycle process in children and youth with and without disabilities [17]. After summarizing the main results, the authors presented a list of suggestions for further interventions, in which they identified the type of the learning bicycle as a key task constraint for acquiring independent cycling. Considering this constraint, the review advises against using the bicycle with lateral training wheels (BTW) and recommends the adoption of an intervention with progressive difficulty that includes the initial use of the balance bicycle (BB). The use of the BTW is one of the most traditional and popular approaches for learning to cycle, which involves attaching two extra lateral training wheels that allow the child to learn how to pedal without the challenge of (im)balance [15]. Despite its popularity, according to the previous review, several authors point out this solution as counterproductive, since children who use it when transitioning to the conventional bicycle (CB) reveal defensive responses of postural rigidity, which hinder and delay the acquisition of independent cycling [13,19,20,21,22,23,24]. The BB consists of a newer learning approach that, by not having pedals nor extra wheels, allows children to explore and train balance from the first moment, without having to, in parallel, control the technique of pedaling, reducing the difficulty of learning the essential function of this mobility tool, as in its original model [25]. On this bicycle, children must propel themselves by pushing their feet on the ground, originating several cycle patterns, i.e., various types of foot support sequences on the ground, such as walking, running, or gliding [26]; this provides a progressive ability to stop contacting the ground during active riding. Perhaps this is the reason why the review suggests that the BB, due to the inherent need to explore body–bicycle stability, may be a more suitable alternative to the BTW [17,27].

Later, this position was corroborated by another retrospective study that investigated the influence of cycle learning paths, i.e., the sequence of the various learning bicycles used, on the age of independent cycling acquisition [15]. Twenty-nine distinct learning paths were identified, which reveals a wide variety of possibilities for learning to cycle, with respect to the task constraint of learning bicycles. It also verified a significant effect of these learning paths in the age of independent cycling acquisition. The learning path that led to the earliest acquisition consisted of using BB followed by CB, with a mean age of 4.16 ± 1.34 years for independent cycling in CB. On the other hand, the more traditional BTW approach followed by CB resulted in a mean age of 5.97 ± 2.16 years, i.e., roughly 1.81 years later than the BB approach [15]. This average learning difference of almost two years represents a huge temporal gap in the child’s or young person’s life, in which they are deprived of the various benefits of cycling.

Children with overweight and obesity often face greater challenges when learning new motor skills, especially those that require balance. Research highlights that excess body weight can impair postural control and stability, slowing down the motor learning process [28,29]. A higher body mass shifts the center of gravity upward and forward, making it more difficult to maintain balance during dynamic activities [30]. Additionally, obesity can induce musculoskeletal changes, such as stiffness and reduced relative muscle strength and functional muscle performance [31,32], thus impairing the ability to adapt to postural challenges [33]. Excess weight also reduces the sensitivity of pressure receptors in areas like the soles of the feet, further complicating balance adjustments during movement [34]. These difficulties, coupled with lower aerobic capacity and reduced resistance to fatigue, often lead children to shy away from physical activity and sports [35]. This avoidance can exacerbate poor body composition, creating a negative cycle that perpetuates inactivity [35,36]. Breaking this cycle is essential. Learning to cycle offers a pathway to transform this negative spiral into a positive one, promoting physical activity and its many benefits for a healthier, more active life [23]. The choice of learning bicycle plays a critical role in addressing these challenges. The BTW limits the immediate need to explore balance, potentially delaying its development. In contrast, the BB encourages children to gradually and independently develop balance at their own pace. For children with overweight or obesity, who face added challenges with balance and fatigue, the BB can provide an ideal environment for controlled balance exploration and fatigue management, offering a more supportive and effective learning experience.

Although previous studies argue that the BB is a more efficient learning tool than the traditional BTW, as it promotes cycling acquisition about two years earlier [15,17,27], according to our review, so far, no study has aimed to investigate and compare which was the most efficient learning bicycle with children with different body compositions. In this sense, considering the urgency and relevance of identifying the best and most efficient strategies to enable our children and youth to cycle independently, including the ones with overweight and obesity, the present study implemented a learning to cycle intervention program, called Learning to Cycle, with the following objectives: (i) to promote learning to cycle and, simultaneously, (ii) to investigate and compare the use of different bicycles during the learning process, i.e., BB and BTW; (iii) to investigate the influence of body composition on this learning process.

## 2. Materials and Methods

### 2.1. Stud Design

The study applied a quasi-experimental design, implementing the Learning to Cycle intervention program, a previously validated program [37]. It included two intervention groups, BB and BTW, and two assessment moments, pre- and post-intervention. The pre-intervention assessment was conducted approximately one week prior to the intervention, including the assessment of independent cycling and other variables that could potentially influence this learning process.

The intervention was divided into two phases: in the first phase, six sessions were conducted with the exclusive use of each group’s learning bicycle, i.e., the children in the BB group only used the BB, while the children in the BTW group only used the BTW; in the second phase, four more sessions were held with the exclusive use of the conventional bicycle (CB) by all children in both groups.

The post-intervention assessment was conducted at the end of each session of the second phase of the intervention, i.e., with the CB, to assess the acquisition of the various cycling milestones and independent cycling. The present study was approved by the Ethics Committee of the Faculty of Human Kinetics of the University of Lisbon (approval number: 22/2019).

### 2.2. Pre-Intervention Assessment

This period covered the assessment of independent cycling, as well as variables potentially influencing the process of learning to cycle, i.e., previous cycling experience [21], motor competence [38], body composition [30], physical activity practice [16].

Considering that this study aimed to compare the process of learning to cycle by using different bicycles, the evaluation of variables that can potentially influence this learning process will allow controlling potential risks of bias, thus ensuring greater internal validity of the study. The assessment protocols are presented below.

#### 2.2.1. Independent Cycling Assessment

In accordance with previous studies [17,37], the independent cycling was defined as the ability to perform, sequentially and without any external help, the following cycling milestones: (i) self-launch, when the cyclist can propel themself and put both feet on the pedals to start pedaling, (ii) ride, when the cyclist can pedal and keep their balance without touching the ground for at least 10 m consecutively and, (iii) brake, when the cyclist is able to brake safely by using the handbrakes and resting both feet on the ground.

Thus, the assessment of independent cycling consisted of evaluating the various cycling milestones described above. To perform this evaluation, children were invited to cycle on a CB, and the researchers observed their behavior and registered their ability to perform each cycling milestone.

#### 2.2.2. Previous Cycling Experience and Physical Activity Practice

Previous bicycle experience (i.e., BB, BTW, bicycle with one lateral training wheel, and CB), and physical activity practice were evaluated through a survey completed by the participants’ parents [37].

#### 2.2.3. Motor Competence

To evaluate motor competence (MC) the Motor Competence Assessment Battery [39] was used. This battery was chosen because it assesses a range of motor skills, which, as per MC’s theoretical construction, are subdivided into three categories: locomotor (i.e., the tests of horizontal jump and shuttle run), stability (i.e., tests lateral jumps and shifting platforms), and manipulative (i.e., tests of throwing, and kicking) proficiency. It is validated with reference values for the Portuguese population. All tests were scored according to the standardized percentiles for the age and sex of Portuguese children. The final MC score was considered as the average of all tests’ percentiles [40].

#### 2.2.4. Body Composition

The body composition of the children was assessed by the Body Mass Index (BMI) and its percentiles, as recommended by the World Health Organization (WHO) [41]. Although we recognize the importance of more advanced measurements, such as fat mass (FM) and lean mass (LM), BMI was chosen considering the sample’s specific characteristics, and due to methodological reasons. The sample included children as young as 3 years old. Measuring skinfolds and circumferences in such young children is extremely invasive, can cause discomfort, and they tend not to be receptive to these procedures. Additionally, according to our knowledge, there are no validated predictive equations for FM and LM for children this young, nor are there validated bioimpedance devices for this age group.

The anthropometric measurements, i.e., weight and height, were performed according to the protocols of the International Society for the Advancement of Kinanthropometry (ISAK) [42], and by an ISAK certified level II anthropometrist.

### 2.3. Intervention Program Learning to Cycle

The Learning to Cycle program consisted of a two-week intervention designed to promote independent cycling learning in preschool and primary school children [37]. The program was developed according to the constraint-led approach (CLA) [43,44] and nonlinear pedagogy (NLP) [45,46]. Both the theoretical frameworks argue that learning is influenced by the various constraints, i.e., individual, task or environment, in which learners are embedded. In this sense, the trainer’s role is not to describe the skill in detail, nor to limit motor solutions, but rather to manipulate the constraints in order to promote variability during learning and thus, consequently, promote the discovery and acquisition of new solutions [46,47]. Based on these premises, all sessions of the program included two components, one of free exploration and the other based on playful learning games. In the free exploration component, the learners could play and explore the bicycle in various spaces, including a training ground and an area of ramps with sand and grass, without any indication or limitation by the researchers. In the game’s component, racing and obstacle playful games were developed and suggested to create instability, promote and guide the exploration of functional and adaptive solutions [46,47,48].

The intervention was conducted by physical education teachers, with a researcher–learner ratio of one to two or, at most, one to three. All the technicians involved in the program were previously selected, considering their motivation to apply a program based on CLA and NLP [49,50], and previous experience in conducting classes with children. The intervention program took place within the school context, with daily sessions of approximately 40 min each, including 10 min of preparation (e.g., individual adjustment of the helmet and bicycle to the anthropometric characteristics of the children), and 30 min of effective practice. As a safety measure and in order to promote the future use of helmets, all children wore helmets in all sessions [51].

### 2.4. Post-Intervention Assessment

The post-intervention assessment consisted of the evaluation of independent cycling ability, according to the cycling milestones explained above. This assessment was conducted at the end of each session of the second intervention phase, i.e., of the four sessions with the CB. Using a daily assessment allowed for the recording of the number of sessions with the CB that each child needed to reach each cycling milestone and independent cycling.

### 2.5. Sample Characterization

Fifty children (23 girls), aged between 3 and 7 years old (M = 5.85 ± 0.93) participated in the study. The children were students from one primary school and two public preschools located in the central region of Portugal. Parental informed consent was obtained, and verbal assent was provided by the children.

Only children who did not previously know how to cycle were included in the study. After the pre-intervention assessment, the participants were divided into the two experimental groups, i.e., BE and BTW, in a stratified random sample according to the variables sex and age.

### 2.6. Data Treatment

The normality of the distribution was tested and assumed only for the variables age, height, and MC for the entire sample. Accordingly, to investigate the differences between groups at the pre-intervention evaluation, the *t*-test was used for the variables age, height, and MC; the U-Mann–Whitney test, for weight and BMI; and, finally, chi-square tests for the variables gender, previous experience, and physical activity practice.

To investigate the differences between groups regarding the number of CB sessions required to acquire cycling milestones and independent cycling, the Mann–Whitney U test was used for between-group comparisons, while the Kruskal–Wallis test with Bonferroni correction was applied for comparisons between body composition groups. For the Mann–Whitney U test, the effect size *r* was calculated, and for the Kruskal–Wallis test, the *η2* effect size was used [52].

To investigate the possible associations between the intrinsic constraints (i.e., BMI, BMI percentile, MC, decimal age) with the number of CB’s sessions needed to acquire the cycling milestones and independent cycling, Spearman’s correlation coefficient was used. A statistical significance level of *p* = 0.05 was adopted for all tests.

## 3. Results

### 3.1. Pre-Intervention Assessment

In the pre-intervention assessment, it was confirmed that all children were unable to perform any of the cycling milestones. The groups did not differ with respect to gender, age, height, weight, BMI, MC, previous experiences on different bicycles, or physical activity practice (all *p_s_* > 0.05). Descriptive statistics by group and for the total sample regarding age, body composition, and MC score are presented below (Table 1).

### 3.2. Intervention Effects on Independent Cycling

As explained in the methods section, the assessment of independent cycling was performed after each of the four sessions with the CB. After the first session with the CB, in the BB group 81% of the children were already able to start, 84% to balance and brake, and 68% to cycle independently. These numbers were lower in the BTW children, since, in this group, after the first session with the CB, only 32% were able to start, 36% to balance, 38.5% to brake, and 16% to cycle independently (Figure 1).

At the end of the intervention, 88% of the children, 45 out of 50, were able to acquire independent cycling (Table 2). It should be noted that all children in the BB group were able to acquire independent cycling, while six children in the BTW group were not. The BB children learned to cycle significantly faster than the BTW children, requiring fewer sessions with CB to self-launch (*U* = 112, *z* = −2.755, *p* = 0.006, *r* = −0.551), ride (*U* = 153, *z* = −2.434, *p* = 0.015, *r* = −0.487), brake (*U* = 150, *z* = −2.739, *p* = 0.006, *r* = −0.548) and independent cycling (*U* = 109.500, *z* = −3.265, *p* = 0.001, *r* = −0.653).

### 3.3. Body Composition and Other Individual Constraints

There was no correlation between the variables decimal age, BMI or MC, and the number of sessions needed to acquire independent cycling (all *p_s_* > 0.05). Regarding the analysis of cycling milestones, only two significant correlations were found, considering the entire sample: BMI and its percentile with the number of sessions required to achieve the ‘ride’ milestone (*r_s_* = 0.467, *p* = 0.002; *r_s_* = 0.458, *p* = 0.002, respectively).

No significant differences were found in the number of sessions required to achieve independent cycling or any cycling milestones when comparing the body composition classification groups (i.e., low weight, normal weight, overweight, and obesity). However, it is noteworthy that children with obesity had the highest mean values for achieving all cycling milestones and for the ability to cycle independently, compared to children in the other body composition groups (Table 3). It is also important to note that, despite the lack of statistical significance, all body composition subgroups (i.e., low weight, normal weight, overweight, and obese) in the BB group required, on average, fewer sessions than their counterparts in the BTW group (Table 4).

## 4. Discussion

The present study implemented the Learning to Cycle program with the following objectives: (i) to promote learning to cycle and, simultaneously; (ii) to investigate and compare the use of different bicycles during the learning process, i.e., BB and BTW; (iii) to investigate the influence of body composition during this learning process. The program revealed a success rate of 88%, since 45 of the 50 participating children learned to cycle independently. This success rate is higher than most studies conducted in the same scope [17,37], showing that it is possible to successfully promote learning to cycle in just two weeks of intervention. It should also be noted that preschool children, as young as three years of age, were included in the study. Under the National Strategy for Active Cycling Mobility 2020–2030 (NSACM) [53], the Portuguese government foresees the inclusion of cycling as an extracurricular subject only from primary school age. However, the present results suggest that this inclusion can be successfully carried out earlier, from preschool. If anticipated, this motor skill stimulation can promote the enculturation of cycling, providing all the potential values and benefits of cycling from very early in life.

Besides the promotion of learning to cycle, an objective successfully accomplished, the present study also aimed to investigate and compare the learning process using the BB and BTW. The results showed that using the BB promotes more efficient learning, i.e., in fewer sessions, than using the BTW (Figure 1). To understand this difference, it is necessary to critically think about the learning process. Learning to ride a bicycle is an important motor milestone in children’s lives [54,55], which involves a complex learning process. To cycle, the child needs to be able to self-launch, pedal, turn, brake, and accomplish all of this while maintaining the stability of both body and bicycle. Thus, to cycle successfully, the child must control and coordinate not only their own degrees of freedom (i.e., motor units, muscles, and joints), but also manage to control and articulate with new degrees of freedom external to themselves (i.e., pedals, steering wheels, and brakes), forming with the bicycle a single stable, functional, and integrated system in the environment in which they move. The BTW is a very popular traditional approach that was initially developed with the goal of simplification, reducing the degrees of freedom and the need for the dynamic search for stability during learning. The coupling of the two extra wheels limits the bicycle’s lateral oscillations and eliminates the balance challenge [56]. In this way, the child first learns how to pedal, and later, when the side wheels are removed, they will struggle with the question of balance while pedaling. The BB promotes a contrary approach. By not having pedals nor extra wheels, the BB is close to the first bicycle model. Through propulsion by feet contact with the ground, delegating to the child the self-management of the degree of stability’s difficulty during displacement, i.e., the greater the number of supports of feet in the ground and the longer the child stays on each support, the greater the stability. It has been observed that children autonomously explore and manage various types and sequences of support, that is, various cycle patterns with the BB. Part of these patterns include flight phases, i.e., phases in which no foot contacts the ground, which may be shorter or longer from running to gliding [26]. The BB thus promotes the exploration of various patterns and, inherently, the exploration of different body–bicycle stability gradients. Therefore, with the BB, the child first learns how to stabilize body and bicycle, with less frequent contact with the ground, and later, when they can glide and the pedal board is introduced, they will incorporate pedaling as well. In an initial analysis, we suggest that the BB, by not having pedals, would also reduce the degrees of freedom involved in learning to cycle, and that this could be the reason for its greater efficiency. However, in a deeper analysis, we can see that the BB promotes more degrees of freedom, since it allows the child to explore more and more varied patterns of cycling in the several planes of movement, not limiting them as BTW does. Could this be the reason for BB’s greater efficiency?

Recently, the hypothesis that learning emerges from a process of dynamic alternations of freezing and freeing the degrees of freedom has been defended within the scope of motor learning of coordinative tasks [57]. According to this view, to promote the learning process it is necessary to disturb the system, in this case, the child–bicycle system. When cycling with the BB, the child is constantly challenged to explore stability with the bicycle. This stimulation may activate the freeze–freeing processes, that is, the locking and unlocking of certain degrees of freedom of their body and the bicycle, and thus allow children to progressively acquire the ability to articulate the stability of their body with the stability of the bicycle. Inversely, when cycling with BTW, children do not experience the challenges of body–bicycle instability. The BTW seems to provide too much stability to the child, not disturbing them, and consequently not activating the freeze–freeing mechanisms necessary for the exploration of new motor solutions and the consequent stability necessary for the use of the CB. Thus, the BTW ceases to be representative of the ability to cycle in the CB [45]. This thesis follows previous studies that consider the use of BTW as a counterproductive approach [13,19,20,21,22,23,24]. According to these authors, children who use BTW when transitioning to CB reveal defensive responses of postural rigidity, freezing upper trunk movements and keeping the upper limbs rigid and inflexible [13,19,20,21,22,23,24]. These defensive responses impair stability control on the bicycle, hindering the learning process of independent cycling [24]. Because these children were not previously compelled to explore the dynamics of body–bicycle stability, when transitioning to CB the perturbations and complexity required are so high that they simply freeze numerous degrees of freedom [19,37]. Although, in the present study, the postural oscillations of the children in the two groups were not investigated and compared, which can be a future line of investigation, all children who did not learn to cycle belonged to the BTW group, and postural rigidity responses in the trunk and upper limbs were anecdotally observed in these children during the transition to the CB, as reported in the previous literature [24]. These results are in line with the only previous study that investigates and compares the methods between these two learning bicycles, i.e., BB and BTW [37]. Although there are several studies investigating this topic, there is a large gap in research analyzing and comparing different learning bikes, and to our knowledge, none of them have simultaneously investigated the effect of body composition during this learning process. This gap in the literature reinforces the relevance and originality of this study.

The reason for the BB’s greater efficiency in learning to cycle thus seems to be linked to the inherent and immediate exploration of stability afforded to children. Ultimately, they can stabilize ourselves (on the bike) without pedaling, but they cannot pedal without being stable. This is in line with previous literature that considers balance acquisition as the most challenging aspect during learning to cycle [27,58,59]. With the BB, children can manage the various degrees of freedom during the learning process, according to their own levels of motor competence and learning pace, i.e., their individual constraints.

The results of the present study thus highlight that there are various ways, and various learning bicycles, that successfully promote learning to cycle. In the BTW group, 19 out of 25 children successfully learned to cycle independently in just two weeks. Nevertheless, the results found support the thesis that the type of learning bicycle used is a task constraint with significant influence on the acquisition of independent cycling, and that the BB provides more efficient learning compared to BTW. In only two sessions with the CB, which corresponds to 60 min of practice, 92% of the children in the BB group learned to cycle independently; on the other hand, only 44% of the children in the BTW group did so. In the next two sessions, the BB group achieved a 100% learning rate, while the BTW group only achieved 76%; the six children in this group who failed to learn to independent cycling during the program would need more practice sessions.

The results of this study show that children with a higher BMI and BMI’s percentile face significant additional difficulties in achieving the ‘ride’ cycle milestone, in which they need to ride the bicycle maintaining their balance. These difficulties can be explained by the relationship between excess body weight and reduced postural stability, as described in the literature. Previous studies highlight that excess weight shifts the center of gravity forward and upwards, complicating the maintenance of balance during dynamic activities such as cycling [30,33]. These findings reinforce the importance of adopting specific strategies to meet the needs of these children, such as the use of the balance bike, which has been shown to be more efficient. Previous studies suggest that BB promotes more efficient acquisition of cycling due to its emphasis on exploring balance patterns from the outset [37,58]. However, children with obesity may need more time to acquire cycling milestones due to factors such as lower resistance to fatigue and changes in proprioception [33,34]. While these difficulties may seem like significant barriers, this study suggests that BB can mitigate some of them by allowing each child to progress at their own pace, gradually exploring instability and better controlling their fatigue. This evidence is corroborated by the lower average number of sessions needed to acquire all the cycling milestones and independent cycling by the obese children in the BB group compared to the BTW group. This finding aligns with the literature that points to variability in practice as an essential factor for motor learning in populations with varied body characteristics [43,44].

It is important to recognize that the small sample size per group, as well as the limited number of overweight and obese children, may have contributed to the lack of statistically significant differences between the learning bikes. This small sample size also restricted the ability to perform more complex statistical analyses, which require a larger sample size and the presence of a normal distribution. This limitation affects the generalizability of the results and underscores the need for future studies with larger and representative samples. Such studies would also enable more thorough statistical analyses to investigate and better understand the complex relationship between body composition, motor competence, and the learning bicycle used. Furthermore, analyzing biomechanical measurements, such as postural oscillations or muscle response during the use of the different bikes, could provide deeper insights into the differences in learning processes, as well as the effect of the individual constraints like body composition.

Despite the limitations, the present study had significant strengths that contribute to its relevance and ecological validity. One of the key strengths was the collection of data in a real-world context, allowing children to learn and practice cycling in an environment that closely mirrors their everyday experiences [60]. This approach minimizes artificial constraints and provides a more accurate representation of how children interact with different types of learning bicycles, enhancing the applicability of the findings to real-life scenarios. This ecological validity is crucial for developing practical recommendations that can be effectively implemented in various educational and recreational settings [61]. Furthermore, the study’s focus on a diverse sample of children, including those with different body compositions, provides valuable insights into how to support all children, particularly those who may face greater challenges in motor learning. Additionally, this study was original in its approach as it is the first, according to our knowledge, to investigate the cycle learning process using different types of learning bicycles, while considering body composition as a variable of interest. This novel perspective provides a foundation for future research and interventions aimed at optimizing cycle learning for children of all body types. By exploring the specific challenges faced by overweight and obese children, this study opened new avenues for targeted strategies that can help these children overcome motor learning difficulties [62], ultimately promoting a healthier and more active lifestyle.

## 5. Conclusions

The results of the present study corroborate that a two-week intervention for learning to cycle can be effective in children from three years old. The results further support the hypothesis that the BB is a more efficient learning bicycle than the BTW. There were significant positive associations between BMI and BMI percentile with the number of sessions needed to achieve the riding milestone, supporting the increased difficulty in balancing for overweight and obese children. Fortunately, there were no significant differences in the acquisition of autonomous cycling among children with different body compositions. Given that BB allows for the exploration of balance from initial contact and at a self-paced rate, it could be a suitable learning bike for children with overweight and obesity.

These findings support the decision to introduce cycling as an extracurricular subject, foreseen in the National Strategy for Active Cycling Mobility 2020–2030, and to suggest its anticipation for preschool education. Our results also support the BB as a preferential learning tool for future interventions in school or out of school contexts. Coaches, teachers, educators, mothers, fathers, and family members should choose the BB for children, from at least three years of age, in order to promote learning to cycle more efficiently and, consequently, incorporating cycling into their way of life from an early age.

## Figures and Tables

**Figure 1 jfmk-10-00053-f001:**
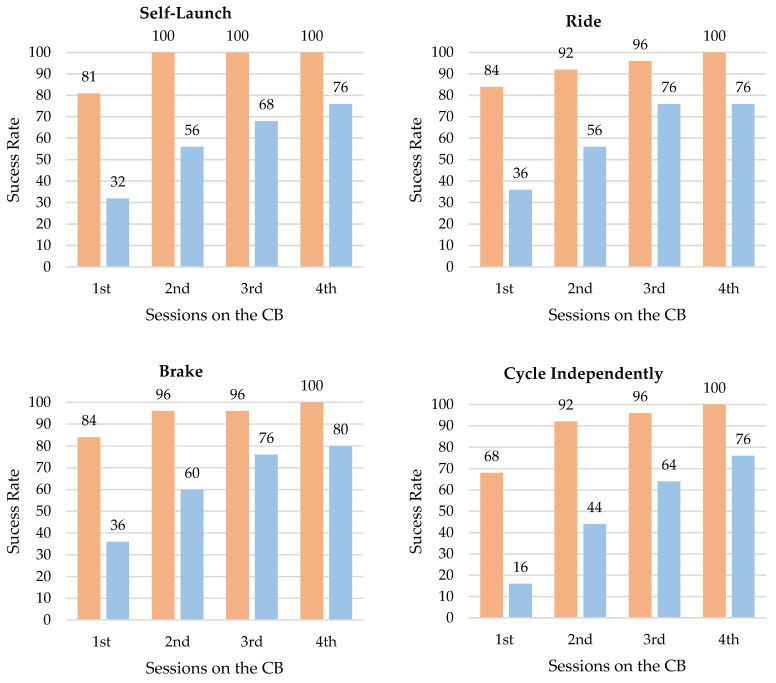
Percentage of success in acquiring cycling milestones and independent cycling by group (BB represented by the orange, and BTW by blue) during the sessions with the conventional bicycle (CB).

**Table 1 jfmk-10-00053-t001:** Descriptive statistics (M, SD, and Mdn) of the variables age, body composition, and CM score, by group and for the total sample.

	BB Group (n = 25)		BTW Group (n = 25)		Total (n = 50)		Comparison Between Groups
	M ± SD	Mdn	M ± SD	Mdn	M ± SD	Mdn	*p*
Age (years)	5.85 ± 0.98	5.81	5.84 ± 0.90	5.68	5.85 ± 0.93	5.73	0.603
Height (m)	1.16 ± 0.09	1.17	1.12 ± 0.06	1.12	1.14 ± 0.08	1.13	0.089
Weight (kg)	22.68 ± 7.18	20.10	21.6 ± 5.10	19.75	22.15 ± 6.21	20.10	0.645
BMI (kg/m^2^)	16.64 ± 3.24	15.93	17.05 ± 3.88	15.87	16.93 ± 3.53	15.93	0.764
BMI’s percentile	57.06 ± 36.12	61.30	60.91 ± 30.98	66.11	58.91 ± 33.45	65	0.726
MC’s score (mean percentile)	46.22 ± 19.39	43.72	41.67 ± 18.09	38.93	43.94 ± 18.69	39.53	0.458
	Percentage		Percentage		Percentage		
Low weight	12%		4%		8%		
Normal weight	56%		72%		64%		
Overweight	12%		0%		6%		
Obesity	20%		24%		22%		

**Table 2 jfmk-10-00053-t002:** Descriptive statistics (M, SD, and Mdn) of sessions with the conventional bicycle needed to acquire cycling milestones and independent cycling, and program’s success rate.

Sessions to Acquire	BB Group		BTW Group		Total		Comparison
	M ± SD	Mdn	M ± SD	Mdn	M ± SD	Mdn	*p* Values
Self-launch	1.19 ± 0.40	1	1.95 ± 1.03	2	1.55 ± 0.85	1	0.006
Ride	1.28 ± 0.74	1	1.79 ± 0.86	2	1.50 ± 0.82	1	0.015
Brake	1.24 ± 0.66	1	1.85 ± 0.93	2	1.51 ± 0.84	1	0.006
Independent cycling	1.44 ± 0.77	1	2.42 ± 1.12	2	1.86 ± 1.05	2	0.001
Success Rate	Percentage		Percentage		Percentage		
	100%		76%		88%		

**Table 3 jfmk-10-00053-t003:** Descriptive statistics (M, SD, and Mdn) of sessions with the conventional bicycle needed to reach cycling milestones and independent cycling, ordered by body composition group for all samples.

Sessions to Acquire	Low Weight		Normal		Overweight		Obesity	
	M ± SD	Mdn	M ± SD	Mdn	M ± SD	Mdn	M ± SD	Mdn
Self-launch	1.67 ± 0.58	2	1.56 ± 0.96	1	1 ± 1	1	1.6 ± 0.70	1.50
Ride	1 ± 1	1	1.45 ± 0.87	1	1 ± 1	1	2 ± 0.71	2
Brake	1 ± 1	1	1.40 ± 0.82	1	1 ± 1	1	1.70 ± 0.82	1.5
Independent cycling	1.67 ± 0.58	2	1.68 ± 0.99	1	1 ± 1	1	2.40 ± 1.17	2

**Table 4 jfmk-10-00053-t004:** Descriptive statistics (M, SD, and Mdn) of sessions with the conventional bicycle needed to reach cycling milestones and independent cycling, ordered by body composition group for each group.

Cycle Milestone	Group	Low Weight		Normal		Overweight		Obesity	
		M ± SD	Mdn	M ± SD	Mdn	M ± SD	Mdn	M ± SD	Mdn
Self-launch	BB	1.67 ± 0.58	2	1 ± 0	1	1 ± 0	1	1.4 ± 0.55	1
	BTW	*	*	2 ± 1.11	2			1.8 ± 0.84	2
Ride	BB	1 ± 1	1	1.21 ± 0.80	1	1 ± 0	1	1.8 ± 0.84	2
	BTW	*	*	1.67 ± 0.9	1			2.25 ± 0.5	2
Brake	BB	1 ± 1	1	1.29 ± 0.83	1	1.33 ± 0.58	1	1.2 ± 0.48	1
	BTW	*	*	1.71 ± 0.99	1			2.17 ± 0.75	2
Independent cycling	BB	1.67 ± 0.58	2	1.29 ± 0.83	1	1.33 ± 0.58	1	1.8 ± 0.84	2
	BTW	*	*	2.21 ± 1.05	2			3 ± 1.22	3

Note: *—missing data as the only child in this group did not acquire any cycling milestone.

## Data Availability

Data availability is possible upon request and with the approval of the institutional ethics committee.

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
