# Peer review of "Learning to Cycle: Body Composition and Balance Challenges in Balance Bikes Versus Training Wheels"

_jfmk, 2025, doi:10.3390/jfmk10010053_

Round 1

Reviewer 1 Report

Comments and Suggestions for Authors

The study analyses the effectiveness of pedal-free bicycles (BBs) compared to bicycles with side wheels (BTWs) in learning to cycle in children, also focusing on difficulties related to high BMI. Although the results suggest that BBs are more effective in promoting rapid learning and balance, there are several methodological and analytical shortcomings.

There was no detailed statistical comparison between the groups at T0, making the claim that the groups were equivalent weak. The assessment of body composition was limited to BMI and its percentiles, excluding more advanced measurements such as fat mass and lean mass, which are essential to investigate the impact of body characteristics on motor performance. The sample size is small, especially in the subgroups with overweight and obesity, limiting the generalisability of the results. The statistical analyses used were insufficient to explore the complex relationships between BMI, motor difficulties and bicycle type. There was also a lack of biomechanical measurements, which would have provided objective data on the motor difficulties and effectiveness of the two bicycle types. The claim that BBs are particularly useful for children with high BMI is not supported by statistically significant differences between body composition groups.

To strengthen the study, it would be necessary to add comprehensive statistical analyses at T0, enlarge the sample and include more detailed assessments of body composition. The adoption of multivariate statistical models and the integration of biomechanical measurements would improve the reliability of the conclusions. Qualitative observations on postural stiffness in BTW children should be systematically validated. In summary, the work represents an interesting contribution, but needs a more rigorous approach to fully support its conclusions.

Author Response

Reviewer 1

General Response: Dear reviewer,

Thank you very much for your comments. You raised several important issues and limitations that deserved our full attention and prompted us to reflect more deeply on past and future methodological decisions. We believe that this review process has significantly contributed to the improvement of the manuscript and will also benefit our future studies. We have made every effort to address all your comments, and our responses are provided below. To make our alterations in the manuscript more visible, we highlighted them by writing in blue.

The study analyses the effectiveness of pedal-free bicycles (BBs) compared to bicycles with side wheels (BTWs) in learning to cycle in children, also focusing on difficulties related to high BMI. Although the results suggest that BBs are more effective in promoting rapid learning and balance, there are several methodological and analytical shortcomings.

There was no detailed statistical comparison between the groups at T0, making the claim that the groups were equivalent weak.

Response: Thank you for your comment. In the “Data Treatment” section, we presented the application of statistical tests at baseline to confirm the absence of significant differences between the groups for the variables under analysis. Additionally, in the “Pre-Intervention Assessment” section, we highlighted the absence of these differences, noting that all p-values were higher than 0.05, the level of significance adopted. However, we recognize that this information could be more emphasized. Therefore, we have added the p-values for each test in Table 1, allowing for a comprehensive review of the descriptive and inferential statistics for both groups.

The assessment of body composition was limited to BMI and its percentiles, excluding more advanced measurements such as fat mass and lean mass, which are essential to investigate the impact of body characteristics on motor performance.

Response: We appreciate the comment and acknowledge the importance of advanced body composition measurements, such as fat mass (FM) and lean mass (LM), to investigate the impact of body characteristics on motor performance. However, we opted for the assessment of BMI and its percentiles as it is a common and scientifically validated practice for evaluating body composition in young children. To assess FM and LM, we could have opted for anthropometric measurements (including skinfolds and circumferences) or bioimpedance analysis. However, our sample included children as young as 3 years old. Measuring skinfolds and circumferences in such young children is extremely invasive, can cause discomfort, and they tend not to be receptive to these procedures. Additionally, to our knowledge, there are no validated predictive equations for FM and LM for children as young as 3 years old, nor are there validated bioimpedance devices for this age group. For these reasons, and recognizing that BMI percentiles are a widely used and accepted method for assessing body composition in young children, we chose to use this method. Recognizing the importance of your comment, we have added to the “Body Composition” section the reasons why we chose this method, as well as identifying its weaknesses and strengths.

WHO.2006.WHO Child Growth Standards: Length/Height-for-age, Weight-for-age, Weight-for-Length, Weight-for-Height and Body Mass Index-for-age: Methods and Development. Geneva: World Health Organization.

The sample size is small, especially in the subgroups with overweight and obesity, limiting the generalisability of the results.

Response: We agree with the reviewer's observation, so we have added it and discussed it in the last paragraph of the discussion. Nevertheless, we also recognize the importance of these studies with smaller samples (and therefore more easily feasible) in exploring the relationship between variables, and whose results (even if not generalizable) indicate and reinforce new directions for future research.

The statistical analyses used were insufficient to explore the complex relationships between BMI, motor difficulties and bicycle type.

Response: We agree with the reviewer. Unfortunately, possibly due to the overall sample size and the size of the subgroups, the percentiles of motor competence and BMI do not exhibit a normal distribution or fulfill the principle of homoscedasticity. Consequently, we were unable to apply linear regression models or analyses of covariance. Acknowledging the significance of such analyses, we have included this limitation in the discussion and suggested it as a direction for future research. We also add Table 4 which shows the average session needed for acquiring each cycling milestone and independent cycling by group and body composition type. This more detailed descriptive data can help to understand this relationship a little better.

There was also a lack of biomechanical measurements, which would have provided objective data on the motor difficulties and effectiveness of the two bicycle types.

Response: Thank you for your comment, which we agree with. We have explored this question as a suggestion for future studies in the last paragraph of the discussion.

The claim that BBs are particularly useful for children with high BMI is not supported by statistically significant differences between body composition groups.

Response: Thank you for your comment. In fact, we did not obtain significant differences, nevertheless, the average data for the acquisition of each cycling milestone by group reinforces that BB can be a suitable option for obese children. With BB, these children took an average of 1.8 sessions to acquire independent cycling, almost half the average of 3 sessions for obese children in the BTW group. This data were not presented, so following your comment we have added Table 4 with them and reflected on them in the discussion.

To strengthen the study, it would be necessary to add comprehensive statistical analyses at T0, enlarge the sample and include more detailed assessments of body composition. The adoption of multivariate statistical models and the integration of biomechanical measurements would improve the reliability of the conclusions. Qualitative observations on postural stiffness in BTW children should be systematically validated. In summary, the work represents an interesting contribution, but needs a more rigorous approach to fully support its conclusions.

Response: Thank you for all your comments and suggestions for improvement, which we welcome. According to your suggestion, we have improved the presentation of the treatment and statistical results for the baseline observation. However, at the moment it is not possible to replicate the entire study by introducing more protocols and evaluating more variables as you suggest. Considering the sample and methodology used, which has a very interesting ecological validity, we have carried out the statistical treatment possible. We reinforce the relevance and importance of these data, despite the fact that the study has limitations and methodological weaknesses, which we humbly acknowledge, present and discuss in the manuscript. To our knowledge, this is the first study to investigate learning to cycle using different learning cycles, analyzing body composition as a variable of interest. In a certain way, this study can be seen as an exploratory study that already gives us interesting and important clues about how to promote more efficient learning to cycle in all children, including those who are overweight or obese and who could therefore have greater difficulty in this motor learning. We greatly appreciate your comments, which prompted us to reflect more deeply. As a result, we have added an additional paragraph to the discussion highlighting the strengths of this study.

Reviewer 2 Report

Comments and Suggestions for Authors

First of all, I thank you for allowing me to conduct this review. Some aspects to be modified:

Objectives should be formulated in the infinitive: “to ...” (ABSTRACT, INTRODUCTION)

Abstract

Include the study design. Add more information about the sample. For example, how many boys and girls.

Introduction

The introduction is quite long compared to the total length of the article. I suggest shortening the introduction, focusing on what is really important about riding a bicycle (remove some of the initial paragraphs, since those explaining the differences between the different types of learning are fine).

I would not ask questions during the introduction. You can include questions at the end of the introduction as a “research question”, but I would not use questions as connectors between paragraphs or as the beginning of paragraphs.

You state the following in the introduction “Additionally, obesity induce musculoskeletal changes, such as stiffness and reduced muscle strength, impair the ability to adapt to postural challenges [33]”. However, some previous research has shown that children and adolescents have even higher muscle strength values than normal-weight children. What kind of strength are they referring to? This should be clarified.

You have already used the abbreviations previously BB or BTW in the introduction, why do you again include the full name at the end of the introduction (lines 125 and 126) instead of using the abbreviations?

The study makes good sense and differs from those conducted in previous literature. The introduction, in general, is well conducted.

Materials and methods

In the design section, the type of study must be specified. It indicates that two semi-experimental groups were used, but I think they mean that it was a quasi-experimental design study. please contrast this.

Further description of the tests used to assess the participants is required.  

Why was this study protocol selected? I mean, 6 sessions of one type, 4 of another type, etc. Was it based on previous research?

More detailed information on the sample, as well as on the calculation of the sample size, is missing. The latter is important in order to know why 50 subjects were sampled.

The non-inclusion of a control group with which to compare the results obtained by each of the groups is an aspect of great relevance for the extrapolation of the results. It should be considered in the statements made in the study, as well as in the limitations of the study.

Results

In Figure 1, include a legend of what each color means. It is true that they put it below in the description, but it is easier to understand if it is in the figure itself.

Table 2 presents the descriptive data, but are there significant differences between the groups? The same in Table 3.

Discussion

Begin with the research objectives (in the infinitive).

The results are not conclusive, therefore, they cannot indicate in the discussion that these differences occur, because statistically they are not proven.

The theoretical justification supporting the discussion with respect to previous research is very limited. They should try to look for justification in previous studies.

In general, the discussion is not poorly written, but there is a lack of information and comparison with previous studies. What does concern me is that assertions are made that are not supported by the results. I need to see a more consistent results section to justify the discussion as written.

Author Response

Reviewer 2

General response:

Thank you very much for your comments. You raised several important issues that deserved our full attention and prompted us to reflect more deeply on past and future methodological decisions. We believe that this review process has significantly contributed to the improvement of the manuscript. We have made every effort to address all your comments, and our responses are provided below. To make our alterations in the manuscript more visible, we highlighted them by writing in green.

First of all, I thank you for allowing me to conduct this review. Some aspects to be modified:

 Objectives should be formulated in the infinitive: “to ...” (ABSTRACT, INTRODUCTION)

Response: Thanks for spotting the mistake, we have already changed the objectives to the infinitive in both the summary and the introduction.

Abstract

Include the study design. Add more information about the sample. For example, how many boys and girls.

Response: Thank you for the comment. As suggested, we have added the type of study and the number of girls to the abstract.

 Introduction

The introduction is quite long compared to the total length of the article. I suggest shortening the introduction, focusing on what is really important about riding a bicycle (remove some of the initial paragraphs, since those explaining the differences between the different types of learning are fine).

Response: Thank you for the suggestion. We have tried to reduce the initial two paragraphs by half, to just one, to allow for some contextualization and reinforcement of the relevance of learning to cycle, while at the same time reducing the size of the introduction. We also have increased the length of the discussion by introducing two paragraphs of considerable length which reflect on the weaknesses and strengths of the present study.

I would not ask questions during the introduction. You can include questions at the end of the introduction as a “research question”, but I would not use questions as connectors between paragraphs or as the beginning of paragraphs.

Response: In line with your suggestion, we have reworded the link between paragraphs by removing the question.

You state the following in the introduction “Additionally, obesity induce musculoskeletal changes, such as stiffness and reduced muscle strength, impair the ability to adapt to postural challenges [33]”. However, some previous research has shown that children and adolescents have even higher muscle strength values than normal-weight children. What kind of strength are they referring to? This should be clarified.

Response: Thank you for your insightful comment. Indeed, some studies have shown that children and adolescents with obesity can exhibit higher absolute muscle strength values compared to their normal-weight peers. However, it is important to distinguish between different types of muscle strength and the contexts in which they are measured. In our statement, we refer to the relative muscle strength and functional muscle performance, particularly in activities that require the support and movement of their own body weight. While obese children may have higher absolute muscle strength, this does not necessarily translate to better functional performance. For example, studies have shown that although obese children may have greater absolute muscle strength in their upper limbs, their lower limb strength and overall functional performance, such as balance and agility, are often compromised due to the additional body weight they must support. We have tried to clarify this issue in the manuscript, in particular by introducing new references.

You have already used the abbreviations previously BB or BTW in the introduction, why do you again include the full name at the end of the introduction (lines 125 and 126) instead of using the abbreviations?

Response: Thank you for noticing. It was an oversight on our part. We have now deleted it and used only the abbreviations.

The study makes good sense and differs from those conducted in previous literature. The introduction, in general, is well conducted.

Response: Thank you very much for your comment. We are pleased that you think the introduction is well-conducted and that the study contains differentiating features.

Materials and methods

In the design section, the type of study must be specified. It indicates that two semi-experimental groups were used, but I think they mean that it was a quasi-experimental design study. please contrast this.

Response: Thank you for your comment. You are correct that the term "semi-experimental" may not be the most accurate description of the study design. The study was indeed a quasi-experimental design, which is characterized by the lack of a control group. In our study, we used two intervention groups (BB and BTW) and conducted pre- and post-intervention assessments to evaluate the outcomes. We clarified it in the manuscript to ensure clarity and accuracy.

Further description of the tests used to assess the participants is required.  

Response: As suggested, we add more information and references to the protocols that had more missing information.

Why was this study protocol selected? I mean, 6 sessions of one type, 4 of another type, etc. Was it based on previous research?

Response: Thank you for your question, we have opted for this protocol as it has already been validated and used previously. We tried to clarify this at the beginning of the “Study Design” section.

More detailed information on the sample, as well as on the calculation of the sample size, is missing. The latter is important in order to know why 50 subjects were sampled.

Response: For this study, we did not perform a sample size calculation. Instead, we aimed to gather the largest sample possible, which culminated in 50 participants. Following your comment (as well as that of another reviewer), we have added more information about the homogeneity of the groups at the initial assessment (highlighted in blue). Additionally, we acknowledge that the sample size and convenience sampling represent limitations of the study, which we have recognized and discussed in the penultimate paragraph of the discussion (also highlighted in blue).

The non-inclusion of a control group with which to compare the results obtained by each of the groups is an aspect of great relevance for the extrapolation of the results. It should be considered in the statements made in the study, as well as in the limitations of the study.

Response: Thank you for your insightful comment. We understand the importance of control groups in experimental designs for extrapolating results. However, in this study, our primary objective was to compare the efficiency of two different types of learning bicycles (BB and BTW) in promoting the acquisition of the independent cycle. Given this comparative nature, both groups needed to be intervention groups, so the comparison between them provided the necessary insights into which bicycle type was more effective. Including a traditional control group, i.e., with no intervention, would not have aligned with our study's goal, which was to determine the relative efficiency of the two bicycle types. Instead, our focus was on understanding which intervention (BB or BTW) better facilitated the learning process. We believe that this approach is appropriate for the research question we aimed to address. Nevertheless, we acknowledge that the absence of a traditional control group is a limitation in terms of generalizability.

Results

In Figure 1, include a legend of what each color means. It is true that they put it below in the description, but it is easier to understand if it is in the figure itself.

Response: Thanks for the suggestion, we have added the color legend for the images in figure format in Figure 1. 

Table 2 presents the descriptive data, but are there significant differences between the groups? The same in Table 3.

Response: Table 2 presents only the descriptive results and the inferential results, including the significant differences, are presented in the text that precedes it. However, to facilitate the reading of the manuscript and following your comment, we added another column in Table 2 with the results of the comparisons. In the case of table 3, as explained in the text that precedes it, there are no significant differences, only two significant correlations whose results are presented in this paragraph.

Discussion

Begin with the research objectives (in the infinitive).

Response: Thank you for the suggestion which we followed. In fact, the discussion should always begin by reminding the reader of the objectives of the study to better introduce the discussion and maintain the internal cohesion of the study.

The results are not conclusive, therefore, they cannot indicate in the discussion that these differences occur, because statistically they are not proven.

Response: We understand and agree with your comment. We sought to clarify in the discussion the lack of statistical significance and to make it clear when we referred to mean values. In line with your comment, we also present and discuss the possible reason for this lack of significance, acknowledging the limitation of our sample with the introduction of a new paragraph (highlighted in blue).

The theoretical justification supporting the discussion with respect to previous research is very limited. They should try to look for justification in previous studies.

Response: Thank you for your comment. Within the available literature, we have tried to improve and strengthen the discussion. However, we would like to reinforce that, according to our research, only one previous study has investigated and compared learning to cycle using these two types of learning bikes, and it did not focus on body composition. This gap in the literature somewhat justifies the lack of further comparisons with previous studies. We have also highlighted this gap and, consequently, the relevance and originality of our manuscript in the discussion.

In general, the discussion is not poorly written, but there is a lack of information and comparison with previous studies. What does concern me is that assertions are made that are not supported by the results. I need to see a more consistent results section to justify the discussion as written.

Response: Thank you for your comment. We have aimed to use more measured language and clarify the absence of significant differences by explicitly stating when we referred to average values. Unfortunately, due to the characteristics of the sample and the results (namely the lack of normality), it was not possible to perform more complex statistical analyses such as regression models. We have added a paragraph in the discussion reflecting on this limitation. Nonetheless, we sought to conduct further statistical analysis, presenting a new Table (Table 4) that highlights the average number of sessions needed to achieve each motor milestone. This table allowed us to identify a clear pattern showing that the BB promotes faster learning (i.e., in fewer sessions) compared to the BTW, even among obese children. Obese children in the BB group took, on average, 1.8 sessions to achieve independent cycling, which is almost half the time compared to the BTW group, which took an average of 3 sessions. Additionally, following your comment and in our effort to improve the discussion, we have added and explored connections to more studies that support the greater efficiency of the BB, emphasizing that, to our knowledge, this was the first study to highlight issues related to body composition. Considering all these improvements, we believe we have enhanced the quality of the manuscript.

Round 2

Reviewer 1 Report

Comments and Suggestions for Authors

The authors fixed all my concerns.

Reviewer 2 Report

Comments and Suggestions for Authors

The authors have responded to all my considerations. The article is ready to be published.